# Establishment of an Efficient Genetic Transformation System in *Sanghuangporus baumii*

**DOI:** 10.3390/jof10020137

**Published:** 2024-02-08

**Authors:** Xutong Wang, Mandi Wang, Jian Sun, Xiaolei Qu, Shixin Wang, Tingting Sun

**Affiliations:** 1Jilin Provincial Key Laboratory of Tree and Grass Genetics and Breeding, College of Forestry and Grassland Science, Jilin Agricultural University, Xincheng Street 2888, Changchun 130118, China; 2College of Forestry, Northeast Forestry University, Hexing Road 26, Xiangfang District, Harbin 150040, China; 3Department of Electrical Engineering, Daqing Normal University, Binxi Road, Daqing 163712, China; 4Department of Food Engineering, Harbin University, Zhongxing Road 109, Nangang District, Harbin 150086, China

**Keywords:** *Sanghuangporus baumii*, genetic transformation, *Agrobacterium tumefaciens*

## Abstract

(1) Background: *Sanghuangporus baumii*, a valuable medicinal fungus, has limited studies on its gene function due to the lack of a genetic transformation system. (2) Methods: This study aimed to establish an efficient *Agrobacterium tumefaciens*-mediated transformation (ATMT) system for *S. baumii.* This study involved cloning the promoter (glyceraldehyde-3-phosphate dehydrogenase, *gpd*) of *S. baumii*, reconstructing the transformation vector, optimizing the treatment of receptor tissues, and inventing a new method for screening positive transformants. (3) Results: The established ATMT system involved replacing the CaMV35S promoter of pCAMBIA-1301 with the *gpd* promoter of *S. baumii* to construct the pCAMBIA-SH-*gpd* transformation vector. The vectors were then transferred to *A. tumefaciens* (EHA105) for infection. This study found that the transformation efficiency was higher in the infection using pCAMBIA-SH-*gpd* vectors than using pCAMBIA-1301 vectors. The mycelia of *S. baumii* were homogenized for 20 s and collected as the genetic transformation receptor. After 20 min of co-culture and 48 h of incubation in 15 mL PDL medium at 25 °C, new colonies grew. (4) Conclusions: These colonies were transferred to PDA medium (hygromycin 4 μg/mL, cefotaxime 300 μg/mL), and the transformation efficiency was determined to be 33.7% using PCR.

## 1. Introduction

*Sanghuangporus baumii* is a well-known medicinal fungus with strong anti-cancer properties [1]. *S. baumii* grows in living or dying trees, most commonly on the tree. *S. baumii* is mainly distributed in northern China, Korea, eastern Russia, and Japan [2]. It has the ability to regulate immunity, inhibit the growth of cancer cells [3], and induce their apoptosis [4]. The main active components of *S. baumii* include polysaccharides, triterpenes, flavonoids, and phenols. Due to these properties, *S. baumii* can be utilized as a valuable cell factory for the production of enzymes and natural products. Therefore, it is important to establish a simple, reliable, and effective genetic modification technology for *S. baumii*. This will facilitate the production of superior strains, enable the study of gene function, and promote the development of gene editing technology.

The genetic transformation technology of fungi involves inserting foreign gene fragments into the genome of the receptor through molecular biology and genetic engineering. This process results in changes to the physiological and biochemical characteristics of the receptor strain through replication, transcription, and translation. Currently, commonly used methods for genetic transformation of fungi include PEG (Polyethylene glycol)-mediated transformation, electroporation transformation, restriction endonuclease-mediated DNA transformation, microprojectile bombardment, and *Agrobacterium tumefaciens*-mediated transformation (ATMT). Among these methods, ATMT is widely favored due to its simplicity, high transformation efficiency, and stable integration of foreign genes into host chromosomes. *A. tumefaciens* cells possess tumor-inducing plasmids (Ti plasmids) that carry transfer DNA (T-DNA). After *A. tumefaciens* infects wounds and enters receptor cells, T-DNA can integrate into genomic DNA [5,6]. Therefore, by inserting target genes into the artificially modified T-DNA region, the transfer and integration of foreign genes into receptor cells can be achieved through *A. tumefaciens* infection. In 1995, the *A. tumefaciens* transformation system was established in *Saccharomyces cerevisiae*, marking the first application of ATMT in fungi. These findings indicated that the T-DNA transfer mechanism in fungi is similar to that in plants, providing a research idea for the future use of ATMT in fungal genetic transformation [7]. Subsequently, De Groot et al. demonstrated that this method can also be applied to filamentous fungi [8]. The genetic transformation system of ATMT was successfully established in *Agaricus bisporus* [9], *Hypholoma sublateritium* [10], *Volvariella volvacea* [11], *Clitopilus passeckerianus* [12], *Ganoderma lucidum* [13], *Lentinula edodes* [14], and others.

The structure of plasmids plays a crucial role in the transformation efficiency as an essential carrier tool in the transformation process. Promoters, in particular, have a significant impact on the expression of screening genes. Chen et al. [15] conducted a study analyzing the effects of different promoters on the transformation efficiency. The results showed that homologous promoters had better effects in *A. bisporus*. In *Pleurotus eryngii*, the CAMV35S promoter successfully activated the expression, resulting in a transformation efficiency of 19% [16]. These findings indicate that different fungi respond differently to various promoters. Therefore, it is crucial to select a promoter with higher efficiency when establishing the genetic transformation system of fungi.

The ATMT process involves several steps, including the collection of young receptor tissues, pre-culturing of *A. tumefaciens*, infecting the *A. tumefaciens*, co-culturing the receptor tissues and *A. tumefaciens* using inducers, and finally screening and detecting the transformants. In 2000, Chen et al. successfully applied ATMT to the genetic transformation of *A. bisporus*, marking the first successful use of this method in edible fungi. When the gill of *A. bisporus* was used as the receptor tissue, the transformation efficiency reached the highest at 64% [15]. Similarly, during the establishment of the *Tricholoma matsutake* genetic transformation system, the transformation efficiency reached 83% when mycelia were used as the receptor tissue [17]. Kilaru et al. [12] discovered that both the mycelia and the gill of *Clitopilus passeckerianus* could serve as receptor tissues for genetic transformation, but the transformation efficiency was higher when mycelia were used. In *Pleurotus ostreatus*, mycelia were considered a better receptor tissue [18], while in *Cordyceps militaris*, successful gene transformation could be achieved using conidia as receptor tissues [19]. When using ATMT to mediate the genetic transformation of fungi, the selection of different receptor tissues and the different developmental stages of the same receptor tissues can significantly affect the genetic transformation efficiency. Generally, the genetic transformation efficiency is higher when using cells or tissues that exhibit strong growth as receptor tissues [20,21]. Therefore, the ideal materials for ATMT are typically young fruiting bodies, gills, spores, and vigorously growing mycelia in fungi. However, obtaining fruiting bodies, spores, and gills from certain fungi can be challenging. As a result, the growing mycelia are commonly used in the establishment of ATMT systems.

In addition to using different fungal organs as receptor tissues, the treatment method of receptor tissues also plays a role in genetic transformation efficiency. *A. tumefaciens* is known to invade organisms through wounds and integrate T-DNA into genomic DNA [22]. Therefore, treating the receptor tissues to increase the number of wounds can significantly improve the transformation efficiency. However, it is important to avoid excessively severe treatment as it may cause the organism to die without obtaining the desired transformants. Therefore, determining how to induce suitable wounds in the receptor tissues is a crucial aspect of this research.

In previous studies, the optimization of the infection time of *A. tumefaciens*, co-culture time, AS concentration, and *A. tumefaciens* strains in the ATMT system was conducted. For the genetic transformation of *Lentinula edodes*, the highest transformation efficiency was achieved when the AS concentration was 200 μmol/L, the *A. tumefaciens* strain was EHA105, and the co-culture time was 48 h [23]. In the case of *Auricularia cornea*, the highest transformation efficiency was observed with the AS concentration of 100 μmol/L and the co-culture time of 48 h [24]. Previous research also demonstrated that for *S. baumii*, the highest transformation efficiency was achieved when the AS concentration was 200 μmol/L, the co-culture time was 48 h, and the *A. tumefaciens* strain was EHA105 [25]. Therefore, this study aims to optimize the genetic transformation vector and receptor tissue treatment for *S. baumii*, rather than focusing on the conventional ATMT conditions. The purpose of this study is to simplify the traditional genetic transformation operation and establish a simple, efficient, and stable genetic transformation system for *S. baumii*. The establishment of this system will facilitate future genetic engineering of *S. baumii*, leading to the improved production of medicinal metabolites and exploration of their synthesis mechanism. Additionally, it will lay the groundwork for establishing a CRISPR system in *S. baumii*.

## 2. Materials and Methods

### 2.1. Cloning of GPD Promoter

*S. baumii* DNA was extracted using the CTAB method. According to the *S. baumii* genome (LNZH00000000.2), the *S. baumii gpd* promoter primer (*gpd*-pro-F: 5′-AAGTGGCTTGAGTTT CGTCGTTGT-3, *gpd*-pro-R: 5′-CACACAGAAAGTAAGCGCACATCG-3′) was designed. The reaction system (50 μL) of PCR contained 25 μL of Premix Taq™ (Takara, Beijing, China), 7.4 μL of ddH2O, 1 μL of Nos-F, 1 μL of Nos-R, and 2 μL of *S. baumii* DNA. The PCR products of *S. baumii gpd* promoters were sequenced by a company (Tsingke, Beijing, China). Then, the PCR products were purified for the subsequent construction of the vector.

### 2.2. Construction of S. baumii Expression Vector

The process of carrier construction was divided into three steps. In the first step, the original pCAMBIA-1301 vectors were double-digested (*Xho* I and *EcoR* I), and then the *Hyg* expression box with a homologous arm was connected to the vector using the Trelief™ SoSoo Cloning Kit (Tsingke, Beijing, China) to obtain the pCAMBIA-*Hyg* vector. In the second step, the vector pCAMBIA-*Hyg* was single-restriction-digested (*EcoR* I), and then the *S. baumii gpd* promoter (SH *gpd* promoter) with a homologous arm was connected to the vector to obtain the pCAMBIA-*gpd* vector. In the third step, the double enzyme cutting (*Hind* III and *Nco* I) was performed on pCAMBIA-*gpd* vector, and SH *gpd* promoter was connected to the vector in the reverse direction to obtain pCAMBIA-SH-*gpd* vector (Figure 1).

### 2.3. Treatment of Receptor Tissues

#### 2.3.1. Preparation of *S. baumii* Mycelia

The *S. baumii* mycelia were inoculated into PDB medium (potato 200 g, glucose 20 g, water 1 L, pH 6.5) and cultured at 180 rpm and 25 °C for 12 days. The liquid spawn was minced for 5 s using a homogenizer (Waring, Stamford, CT, USA) under sterile conditions [26]. The broken mycelia were inoculated into PDB medium at 25 °C for 8 days. The cultured mycelia were homogenized for 30 s with a homogenizer. The homogenized liquid was placed in a 50 mL centrifuge tube for 6000× *g* for 10 min; then, the supernatant was discarded. The precipitated mycelia were collected for the genetic transformation of *S. baumii*.

#### 2.3.2. Preparation of *S. baumii* Flakes

The *S. baumii* mycelia were inoculated into PDA medium (potato 200 g, glucose 20 g, agar 17 g, water 1 L, pH 6.5) at 25 °C for 8 days. The growing edges of the mycelia are cut into small round flakes with a diameter of 4 mm using a hole punch. These flakes were then used as receptor tissues for the genetic transformation of *S. baumii*.

#### 2.3.3. Preparation of *S. baumii* Protoplast

The *S. baumii* mycelia were inoculated into PDB medium and incubated at 25 °C for 5 days to obtain the aerial mycelium. Then, 10 mL of lysozyme solution (0.2 g of lysozyme was dissolved in 10 mL of 0.6 moL/L mannitol solution) was added into 1 g of *S. baumii* aerial mycelium and enzymolized at 30 °C for 3 h. The enzymatic hydrolysate was filtered using a G2 sand funnel in order to remove the incomplete enzymolysis mycelia. The filtrate was centrifuged for 10 min to 5000× *g*, and then the supernatant was discarded [27]. The precipitates were dissolved in mannitol solution (0.6 moL/L); then, the resuspension solution was centrifuged again for 10 min, and the supernatant was discarded. The precipitate was resuspended with 1–3 mL mannitol solution to obtain *S. baumii* protoplasts.

### 2.4. Sensitivity Test for Hygromycin in the Mycelium of S. baumii Receptor Tissues

#### 2.4.1. Hygromycin Resistance of *S. baumii* Mycelia

A total of 0.2 g of *S. baumii* mycelia was resuspended with 1 mL IM medium (K-Buffer 1 mL, M-N solution 400 μL, 1% CaCl_2_ 100 μL, 20% NH_4_NO_3_ 250 μL, 0.01% FeSO_4_ 1 mL, 0.05% ZnSO_4_∙7H_2_O 100 μL, 0.05% MnSO_4_∙H_2_O 100 μL, 0.05% CuSO_4_∙5H_2_O 100 μL, 0.05% Na_2_MoO_4_∙2H_2_O 100 μL, 0.05% H_3_BO_3_ 100 μL, 2 mol/L glucose 500 μL, 50% glycerinum 1 mL, 1 mol/L 4-morpholinoethanesulfonic acid 4 mL, 100 μmol/mL acetosyringone 200 μL, 50% tween-80 20 μL, distilled water 91.03 mL); then, 200 μL of the resuspended solution was smeared on Co-IM medium (K-Buffer 1 mL, M-N solution 400 μL, 1% CaCl_2_ 100 μL, 20% NH_4_NO_3_ 250 μL, 0.01% FeSO_4_ 1 mL, 0.05% ZnSO_4_∙7H_2_O 100 μL, 0.05% MnSO_4_∙H_2_O 100 μL, 0.05% CuSO_4_∙5H_2_O 100 μL, 0.05% Na_2_MoO_4_∙2H_2_O 100 μL, 0.05% H_3_BO_3_ 100 μL, 2 mol/L glucose 250 μL, 50% glycerin 1 mL, 1 mol/L 4-morpholinoethanesulfonic acid 4 mL, 100 μmol/mL acetosyringone 200 μL, 50% tween-80 20 μL, agar 1.6g, distilled water 91.28 mL). After 48 h, the Co-IM system was coated with PDL (potato 200 g, glucose 20 g, low metlin agarose 5 g, water 1 L, pH 6.5) containing hygromycin (0 μg/mL, 2 μg/mL, 4 μg/mL, 6 μg/mL, 8 μg/mL, 10 μg/mL). Then, the medium was cultured at 25 °C, and the growth of mycelia was observed in order to determine the most suitable concentration of hygromycin for the *S. baumii*.

#### 2.4.2. Hygromycin Resistance of *S. baumii* Flakes

*S. baumii* flakes were inoculated into PDA medium with hygromycin. The medium was then cultured at 25 °C, and the growth of mycelium was observed.

#### 2.4.3. Hygromycin resistance of *S. baumii* protoplast

*S. baumii* protoplasts were coated in regeneration medium (mannitol 109.3 g, maltose 10 g, yeast powder 4 g, glucose 4 g, agar 15 g, distilled water 1 L, pH 6.5) containing hygromycin. The medium was then kept at 25 °C for 10–15 days, and the protoplast regeneration was observed.

### 2.5. Infection and Co-Culture

A. tumefaciens EH105 (Weidi, Shanghai, China) containing the vector (pCAMBIA-1301, pCAMBIA-SH-gpd) was inoculated on LB medium (kanamycin sulfate 50 μg/mL, rifampicin 20 μg/mL) and cultured at 28 °C for 2 days. The single colonies were then inoculated into LB liquid medium (kanamycin sulfate 50 μg/mL, rifampicin 20 μg/mL) and cultured in a constant-temperature oscillating incubator at 28 °C with a speed of 180 r/min until OD600 = 0.4–0.6 was reached. The bacterial solution of A. tumefaciens was centrifuged for 10 min at 6000× *g*, and the supernatant was discarded. The precipitate was then resuspended with IM medium to reach OD600 = 0.6–0.7. The receptor tissues were mixed with the infective solution of A. tumefaciens and incubated for 20 min at 28 °C. The infected receptor tissues were subsequently cultured on Co-IM medium at 25 °C for 48 h.

### 2.6. Screening for Positive Transformants

The infected mycelium and protoplasts of *S. baumii* were covered with 15 mL PDL medium (hygromycin 4 μg/mL, cefotaxime 300 μg/mL) and cultured at 25 °C until new mycelium grew. The *S. baumii* flakes were transferred to PDA medium (hygromycin 4 μg/mL, cefotaxime 300 μg/mL) using sterile tweezers at 25 °C until new mycelium grew. Then, the new mycelium was transferred to PDA medium (hygromycin 4 μg/mL). After two rounds of hygromycin screening, the new mycelium was cultured in PDA medium without antibiotics for PCR detection to identify positive transformants. The mycelium DNA of wild type and transformants was extracted using a T5 Direct PCR Kit (Tsingke, Beijing, China). The *Nos*-F (5′-GATCGTTCAAACATTTGGCAATA-3′) and *Nos*-R (5′-GATCTAGTAACATAGATGACACCG-3′) primers were synthesized based on the *Nos* terminator. The reaction system (20 μL) contained 10 μL of 2 × T5 Direct PCR Mix, 7.4 μL of ddH_2_O, 0.8 μL of *Nos*-F, 0.8 μL of *Nos*-R, and 1 μL of *S. baumii* cDNA. To improve the accuracy of screening, another set of primers (*Hyg*-F: 5′-CTACAAAGATCGTTATGTTTA-3′, *Hyg*-R: 5’-GTCTGCTGCTCCATACAAG-3′) were designed to validate the transformants. Thermal cycling was performed at 98 °C for 3 min, followed by 35 cycles at 98 °C for 10 s, annealing temperature for 30 s, 72 °C for 40 s, and a final extension at 72 °C for 5 min. The results were counted, and the transformation efficiency was calculated. The transformation efficiency was the amount of positive mycelium mass divided by the amount of germinated mycelium mass.

### 2.7. Optimization of ATMT System for S. baumii

First, pCAMBIA-1301 and pCAMBIA-SH-*gpd* vectors were transferred into *S. baumii*. Positive transformants were screened to calculate the transformation efficiency. The priming function of the *S. baumii gpd* promoter was verified, and the genetic transformation of the binary vector for *S. baumii* was screened. Then, the receptor tissues were processed in different ways (mycelia, flakes, and protoplasts). The transformation efficiency of the three methods was calculated in order to select the best treatment method for the receptor. Finally, the degree of treatment of the receptor tissues was optimized, and an efficient and simple ATMT system of *S. baumii* was established. Each experiment consisted of three replicated groups.

## 3. Results

### 3.1. Effect of Hygromycin on Growth of S. baumii

The hygromycin sensitivity test was crucial for screening positive transformants. The procedure could be simplified by obtaining the appropriate hygromycin concentration for the receptor tissues. *S. baumii* mycelia were placed in the medium containing hygromycin (0 μg/mL, 2 μg/mL, 4 μg/mL, 6 μg/mL, 8 μg/mL, 10 μg/mL). When the concentration of hygromycin was 4 μg/mL, the growth of mycelia could be inhibited exactly, but the mycelia did not die (Figure 2).

*S. baumii* flakes were inoculated with PDA medium containing hygromycin (0 μg/mL, 2 μg/mL, 4 μg/mL, 6 μg/mL). Similarly, when the concentration of hygromycin was 4 μg/mL, flakes and protoplast growth were inhibited exactly (Figure 3). 

### 3.2. Effect of Promoter on Transformation Efficiency in S. baumii

Both pCAMBIA-1301 and pCAMBIA-SH-*gpd* vectors can activate gene expression in *S. baumii*, but the transformation efficiency was different. After hygromycin screening, mycelium masses were selected for PCR validation. The transformation efficiency of the system using pCAMBIA-1301 and pCAMBIA-SH-*gpd* vectors was 10.8% and 22.6%, respectively. Therefore, the homologous *gpd* promoter was more suitable for the genetic transformation of *S. baumii* (Figure 4).

### 3.3. Effect of Treatment of Receptor Tissues on Transformation Efficiency in S. baumii

The treatment method of the receptor tissues can affect the transformation efficiency. In addition, contamination and simplicity are also key factors to be considered. There are three treatment methods for receptor tissues of *S. baumii*: mycelia, flakes, and protoplasts (Figure 5). Their transformation efficiency was 30.6%, 20.8%, and 30.4%, respectively. The contamination of the transformed material had a great relationship with the operation steps and the regeneration efficiency of the material. The preparation of protoplasts was complicated, and the regeneration efficiency of the protoplasts was slow.

In this study, there was material contamination in the protoplast treatment, and the material was not contaminated by using the other two treatments. The preparation of mycelia was the fastest and easiest.

### 3.4. Effect of Homogenization Time of S. baumii Mycelia on Transformation Efficiency

In order to improve the transformation efficiency, the homogeneity time of mycelia was studied during the genetic transformation process. The homogenization time of mycelia was set to 10 s, 20 s, and 30 s. Then, the mycelia were infected and co-cultured with *A. tumefaciens*. The screening results show that, when the time was 20 s, the transformation efficiency was the highest (33.7%). 

## 4. Discussion

*S. baumii* is a rare and traditional medicinal fungus in China known for its various functions, including anti-tumor and immune regulation. However, the research progress on the molecular mechanism of medicinal metabolites in *S. baumii* is currently slow. One contributing factor to this phenomenon is the lack of a stable and efficient genetic transformation system in *S. baumii*. ATMT is a commonly used technique for genetic transformation, allowing for the integration of stable single-copy genes into the genome [26]. However, the commercial vectors available for ATMT systems are mostly designed for plants and utilize plant-specific strong promoters (CaMV35S promoters). The activation efficiency of the CaMV35S promoter in fungi is relatively low, which negatively affects the transformation rate [28,29]. Previous studies have not investigated whether the SH *gpd* promoter is capable of initiating gene expression. Therefore, the SH *gpd* promoter sequence was analyzed using software. The analysis revealed the presence of transcription initiation sites, suggesting that the SH *gpd* promoter might indeed have the ability to initiate gene expression. Furthermore, the sequence also contained several core functional elements. However, it is important to note that the bioinformatics study conducted was purely predictive in nature, and further experimental verification is required to fully comprehend the promoter’s function. 

In this study, the pCAMBIA-1301 vector was modified by replacing the plant promoters with fungal homologous promoters, specifically using the SH *gpd* promoter. This modification resulted in the production of a new binary vector for the genetic transformation of *S. baumii*. The use of fungal homologous promoters to replace plant promoters is a commonly employed method, as demonstrated in previous studies involving *L. edodes* [26], *G. lucidum* [13], and *Agaricus bisporus* [30], which showed improved transformation efficiency. The *gpd* promoter region, in particular, has been proven to be highly efficient in directing the expression of heterologous genes in *S. cerevisiae* [31]. Additionally, *gpd* promoter sequences have been utilized as promoters in vectors designed for the transformation of filamentous fungi such as *Aspergillus nidulans* [32], *Schizophyllum commune* [33], and *Agaricus bisporus* [34]. In this study, the pCAMBIA-1301 and pCAMBIA-SH-gpd vectors were introduced into *S. baumii* using the ATMT method. The transformation efficiency using the SH *gpd* promoter (22.6%) was higher than that using the CaMV35S promoter (10.8%). This demonstrates that the SH *gpd* promoter effectively initiated gene expression in *S. baumii*. While both the CaMV35S and SH *gpd* promoters could activate gene expression, it was evident that homologous promoters are more efficient.

The bivector p1301-SH-*gpd* used in this study carried a screening label of hydomycin resistance. The tolerance of hygromycin is related to many factors, such as species, receptor tissues, medium material, etc. The hygromycin tolerance of *L. edodes* mycelia cultured in PDA medium was 5 μg/mL [29]. The hygromycin tolerance of protoplasts of *G. lucidum* cultured in YPD medium was 100 μg/mL [27], and the hygromycin tolerance of *F. velutipes* mycelia cultured in YPD medium was 9 μg/mL [35]. Therefore, when using hygromycin as a resistance screening marker, it is necessary to test the tolerance of the receptor to hygromycin. In this study, the tolerance of hygromycin could be influenced by various factors, including species, receptor tissues, and medium material. For instance, *L. edodes* mycelia cultured in PDA medium exhibited a hygromycin tolerance of 5 μg/mL [29]. On the other hand, protoplasts of *G. lucidum* cultured in YPD medium showed a hygromycin tolerance of 100 μg/mL [27], while *F. velutipes* mycelia cultured in YPD medium had a tolerance of 9 μg/mL [35]. In our study, we investigated the tolerance of *S. baumii* to hygromycin and determined that a concentration of 4 μg/mL was sufficient to inhibit the growth of *S. baumii* without causing its death. In our previous studies, only the appropriate hygromycin concentration was indicated, but the screening process and pictures were not listed [36,37]. In this paper, the screening for the hygromycin tolerance of mycelium was explained in detail.

Currently, protoplasts, mycelia, or spores are commonly used as receptor tissues for the genetic transformation of fungi. Protoplasts are theoretically more favorable for genetic transformation [37], but they are more challenging to prepare and regenerate. While protoplasts often yield a high transformation efficiency, their use as receptor tissues is associated with higher costs due to the need for lysozyme. Spores are also a viable option for genetic transformation, but it takes a long time for *S. baumii* to mature, so it is difficult to collect spores. As a result, the flakes and mycelia of *S. baumii* are used as receptor tissues instead, eliminating the need for lysozyme treatment to avoid causing wounds. The mycelia treatment, which involved homogenization using a homogenizer, proved to be more efficient compared to the flake treatment. The use of a homogenizer for genetic transformation represents an innovative aspect of our research. In our past studies, Wang et al. [36] only used this method to complete the overexpression of *FPS* of *S. baumii* and did not optimize the genetic transformation procedure. After that, Liu et al. [37] only optimized the strains of *A. tumefaciens*, concentrations of acetosyringone, and lengths of co-culture duration of the genetic transformation system of *S. baumii*. In this study, the concentration of hygromycin, the promoter of transformed vectors, and treatment of receptor tissues were optimized, which was very effective in simplifying the procedure and improving the efficiency (33.7%).

And we also made innovations in positive screening. In most fungi, the flakes or protoplasts are transferred onto a new medium at the end of co-culture. In this study, the culture surface was flat because the mycelia were homogenized. After co-culture, PDL medium was directly coated on the mycelia. This improvement made the ATMT process much easier, requiring only one step to screen out most false positives in a Petri dish. Genetic transformation is a painstaking experiment, and improvements in the method can reduce the workload of scientific researchers. This method can be used not only in *S. baumii* but also in the establishment of other fungal genetic transformation systems. There was no mention of this technological innovation in any of our previous studies.

The time of mycelia homogenization was related to the degree of mycelia injury. When the homogenization time was longer, the mycelia were more damaged, which is unfavorable to the transformation of mycelia. Therefore, it was very important to find the appropriate homogenization time for the transformation of *S. baumii*. The transformation efficiency was highest when the homogeneous time was 20 s. With the extension of the homogenization time, the transformation efficiency decreased gradually. This phenomenon may be because the mycelia were too damaged to withstand *A. tumefaciens* infection.

The specific steps were as follows: the mycelia of *S. baumii* were homogenized for 20 s using a homogenizer and then centrifuged at 6000× *g* for 10 min to collect the precipitated mycelia. The *A. tumefaciens* infection lasted for 20 min. After co-culturing for 48 h, it was covered with 15 mL of PDL medium (hygromycin 4μg/mL, ceftiofur 300 μg/mL) and incubated at 25 °C for 5 to 10 days to allow new colonies to grow (Figure 6). The new colonies were transferred to a new resistant PDA medium (hygromycin 4 μg/mL, ceftiofur sodium 300 μg/mL), and DNA was extracted for PCR verification. The transformation efficiency of this ATMT system for *S. baumii* was 33.7%. In addition, we had successfully achieved the overexpression of *S. baumii* genes by using this method [36,37].

## 5. Conclusions

In this study, we optimized the genetic transformation system of *S. baumii* by cloning the *gpd* promoter, constructing the vector, testing *S. baumii*’s tolerance to hygromycin, verifying the function of the gpd promoter, and processing the receptor tissues. For the first time, this study successfully optimized the steps involved in the process. Notably, we innovatively utilized a homogenizer to handle the receptors and employed PDL medium for screening, enabling the rapid acquisition of transformers. These findings provide a valuable tool for future research on gene function and gene editing and lay the foundation for establishing the RNAi and CRISPR systems of *S. baumii*.

## Figures and Tables

**Figure 1 jof-10-00137-f001:**
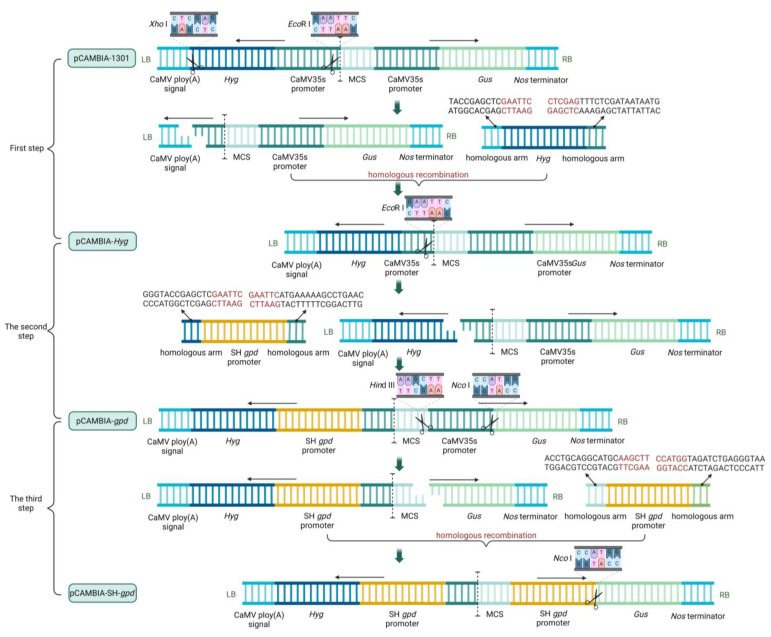
Blueprint of carrier construction.

**Figure 2 jof-10-00137-f002:**
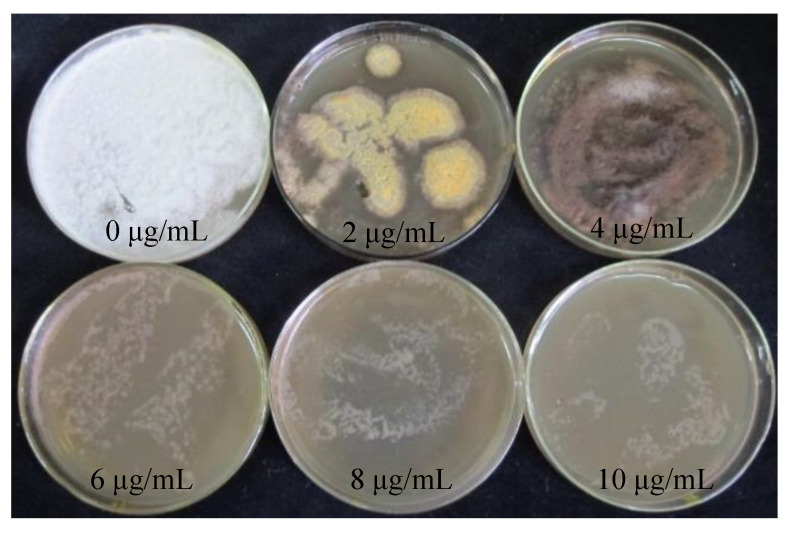
Sensitivity of *S. baumii* mycelia to hygromycin.

**Figure 3 jof-10-00137-f003:**
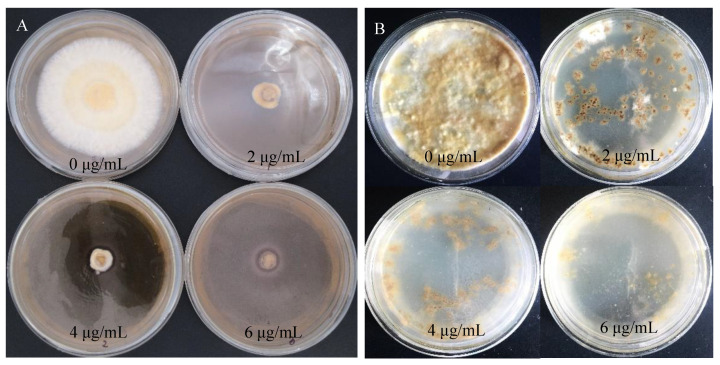
Sensitivity of *S. baumii* to hygromycin. (**A**) Sensitivity of *S. baumii* flakes to hygromycin. (**B**) Sensitivity of *S. baumii* protoplasts to hygromycin.

**Figure 4 jof-10-00137-f004:**
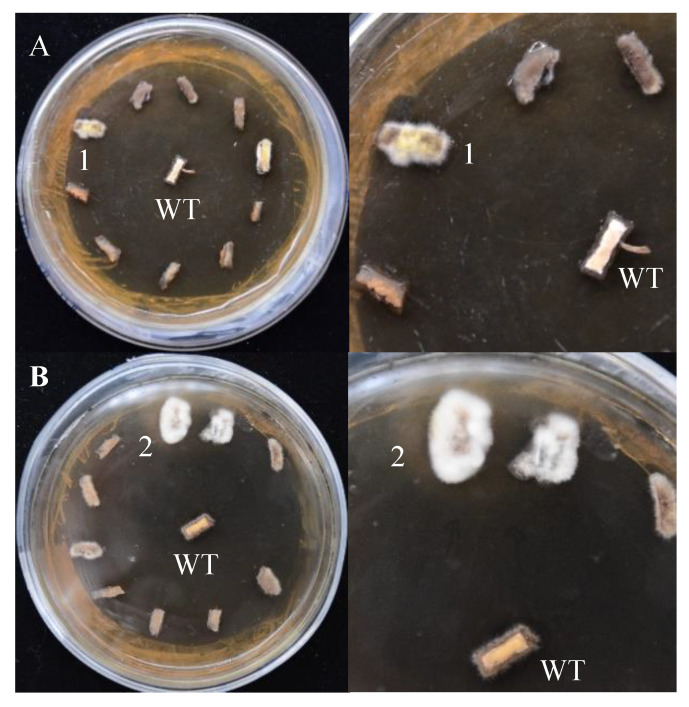
The screening results of positive transformants (on the right is the detailed diagram). (**A**) pCAMBIA-1301 vectors. (**B**) pCAMBIA-SH-*gpd* vectors. WT: the mycelium mass was not co-cultured with *A. tumefaciens*. 1 and 2: transformants 1 and 2.

**Figure 5 jof-10-00137-f005:**
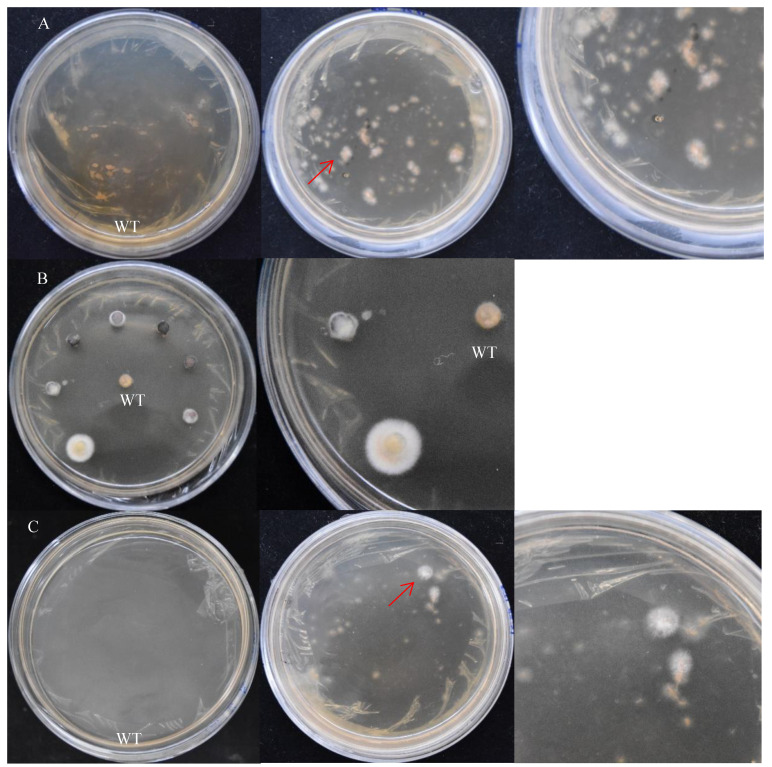
The screening results of positive transformants (on the right is the detailed diagram). (**A**) Mycelia treatment. (**B**) Flake treatment. (**C**) Protoplast treatment. WT: the mycelium mass was not co-cultured with *A. tumefaciens*. The red arrow points to the transformants.

**Figure 6 jof-10-00137-f006:**
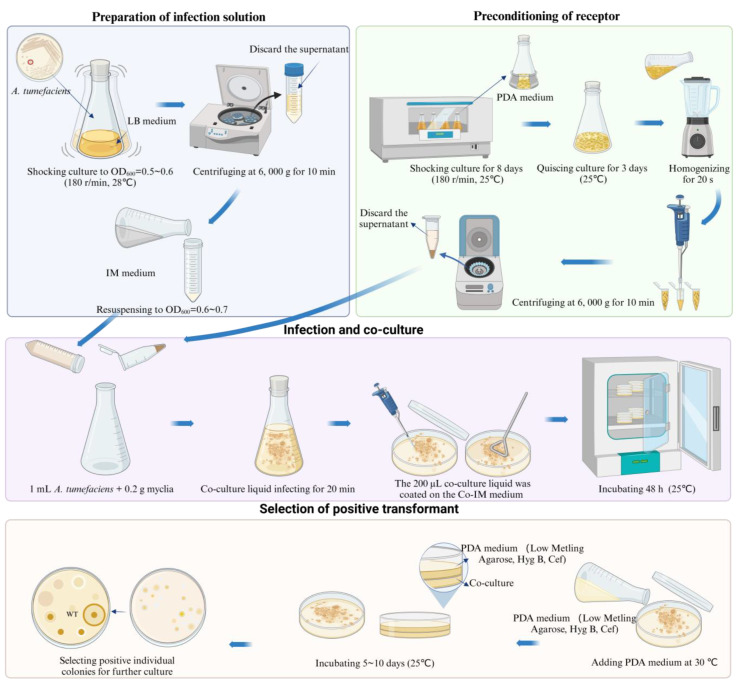
Procedure of genetic transformation of *S. baumii*.

## Data Availability

Data are contained within the article.

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
