# Peer review of "Establishment of an Efficient Genetic Transformation System in Sanghuangporus baumii"

_jof, 2024, doi:10.3390/jof10020137_

Round 1

Reviewer 1 Report

Comments and Suggestions for Authors

Line 24:  hygromycin resistant:  DO YOU MEAN  Hygromycin containing? I am not sure I understand

Line 272: Only 0???  What does this mean?

Line 310: even higher, at 22.7% -  HIGHER THAN WHAT? What do you want to stress here?

Line 365:  or cannot accept foreign genes:  FAR TOO VAGUE> PLEASE EXPLAIN

Line 473: ref. 37 – 2023, Microbiology 51(3). 1. Microbiology 2023 Volume is  #92 not 51!   2.   Issue 3 does not contain a paper by Liu, Liu & Zou. WHERE CAN I FIND THIS PAPER?

Line 477: WHAT IS THIS?

Comments on the Quality of English Language

 English is poor

Reviewer 2 Report

Comments and Suggestions for Authors

The ms by Wang et al describes improvements to the method for Agrobacterium-mediated transformation of the fungus Sanghuangporus baumii, which was published last year by the          same group. Reading the article it did not become immediately clear which steps were improved as the focus was sometimes on irrelevant other points, and descriptions of the outcome of experiments were rather limited. Below I give more detailed feedback.

Line 17 cloning the promoter of S.b. Of course there are numerous promoters in S.b. It is better to mention immediately that you have clonesd the gpd promoter of S.b. to compare its use in comparison to the CaMV 35S promoter

Line 19 what are converters?

Line 25 and line 253 and throughout what does it mean, a transformation frequency of 33.7%? What was measured, the growth of mycelium segments on selective medium after AMT? Also, and even more importantly, how was this calculated? How many mycelia were used in the co-cultivations and how many times were the experiments repeated to be able to calculate and compare numbers and assess their significance?

Line 43 DMNA

Line 130 why is the S.b. gpd promoter called the SH gpd promoter?

Line 134 Fig1 and also Fig 6 are unclear and need to be improved

Line 138, 146 what is S.b. spawn?

Line 195 is Co-IM the same as IM?

Line 222 and whole paragraph 3.1 should be removed: this bioinformatics analysis is meaningless: presence of such small (mostly plant-specific) boxes in the promoter areas does not mean anything biologically. Most of them will be present numerous times in the genome. The only way to find out is to do experiments with promoter mutations in these regions, or perhaps by direct comparison with other fungal promoters for which such experiments have been done.

Line 237 coated?

Line 255 Figure4 what does this show? Are these 10 mycelia that had been co-cultivated with Agro and in the middle a segment called wt that has not been co-cultivated? Please give better explanation in the legend. Then in the text describe the number of independent experiments that were done in this way and how many plates were done in each experiment and then the results of each experiment and the calculation of the average transformation frequency for use to compare the influence of the variables tested.

Line 261 pollution rate?

Line 263 contamination of transformation material?

Line 271 rate

Line 314 bivector? Hydomycin

Line 379 the authors suggest that S.b. was now transformed for the first time, but they already published last year about it (reference 37 in this ms).

Comments on the Quality of English Language

English in general is not too bad. Sometimes the wrong words are used as I have indicated specifically above.

Author Response

For Technical Note

Establishment of an Efficient Genetic Transformation System in Sanghuangporus baumii

Response to Reviewer 2 Comments

1. Summary

Thank you very much for taking the time to review this manuscript. Your comments and questions are very important to our manuscript. We revised our manuscript in detail. Your opinion is very clear and meaningful. We quite agree with your point of view, and have made changes according to each comment. I hope our manuscript can get a good result. If there are still some unclear, please be sure to make suggestions for us.

2. Questions for General Evaluation

Reviewer’s Evaluation

Response and Revisions

Does the introduction provide sufficient background and include all relevant references?

Yes/Can be improved/Must be improved/Not applicable

We have given detailed modifications in Section 3.

Are all the cited references relevant to the research?

Yes/Can be improved/Must be improved/Not applicable

Is the research design appropriate?

Yes/Can be improved/Must be improved/Not applicable

Are the methods adequately described?

Yes/Can be improved/Must be improved/Not applicable

Are the results clearly presented?

Yes/Can be improved/Must be improved/Not applicable

Are the conclusions supported by the results?

Yes/Can be improved/Must be improved/Not applicable

3. Point-by-point response to Comments and Suggestions for Authors

The ms by Wang et al describes improvements to the method for Agrobacterium-mediated transformation of the fungus Sanghuangporus baumii, which was published last year by the same group. Reading the article it did not become immediately clear which steps were improved as the focus was sometimes on irrelevant other points, and descriptions of the outcome of experiments were rather limited. Below I give more detailed feedback.

Comments 1: Line 17 cloning the promoter of S.b. Of course there are numerous promoters in S.b. It is better to mention immediately that you have clonesd the gpd promoter of S.b. to compare its use in comparison to the CaMV 35S promoter.

Response 1: Thank you for pointing this out. We agree with this comment. Therefore, we have revised the description here.

Line 17: The study involved cloning the promoter (glyceraldehyde-3-phosphate dehydrogenase, gpd) of S. baumii,

Line 22-23: The study found that the transformation efficiency was higher in the infection using pCAMBIA-SH-gpd vectors than using pCAMBIA-1301 vectors.

Comments 2: Line 19 what are converters?

Response 2: We are very sorry. Converters may be inaccurate. So we changed the word. Thank you for pointing this out.

Line 19: positive transformants

Comments 3: Line 25 and line 253 and throughout what does it mean, a transformation frequency of 33.3%? What was measured, the growth of mycelium segments on selective medium after AMT? Also, and even more importantly, how was this calculated? How many mycelia were used in the co-cultivations and how many times were the experiments repeated to be able to calculate and compare numbers and assess their significance?

Response 3: We quite agree with you. We have not considered before, and we have added calculation methods to the manuscript.

Line219: The transformation efficiency was number of positive mycelium mass divided by number of germinated mycelium mass.

The transformed mycelia after infection and co-culture can grow on the selected medium, while the non-infected mycelia cannot grow. In Fig. 5, WT represents the non-infected mycelia.

We removed the germinated mycelium mass and placed them in a new selected medium. The re-germinated mycelium in a new selected medium. The number of positive mycelium pieces was obtained by PCR verification. The mycelium used in co-culture could not be accurately calculated. However, in each experiment, we selected more than 3 mycelium mass, and each experiment consisted of three replicated groups. The final transformation rate is the average of the three experiments.

Comments 4: Line 43 DMNA

Response 4: We are very sorry, here is a typo.

Line 48: DNA

Comments 5: Line 130 why is the S.b. gpd promoter called the SH gpd promoter?

Response 5: This is a name based on Chinese. We have always used this name, so we also use this name in this manuscript.

Comments 6: Line 134 Fig1 and also Fig 6 are unclear and need to be improved

Response 6: Thank you for your suggestion. We have made improvements. Now you can zoom in to see these images.

Comments 7: Line 138, 146 what is S.b. spawn?

Response 7: It's more appropriate to use mycelium here, which we've modified.

Line 143, 146: mycelia

Comments 8: Line 195 is Co-IM the same as IM?

Response 8: No, there was mention of their formula in 167-179. The main difference with Co-IM was that the glucose content was halved and agar is added.

Comments 9: Line 222 and whole paragraph 3.1 should be removed: this bioinformatics analysis is meaningless: presence of such small (mostly plant-specific) boxes in the promoter areas does not mean anything biologically. Most of them will be present numerous times in the genome. The only way to find out is to do experiments with promoter mutations in these regions, or perhaps by direct comparison with other fungal promoters for which such experiments have been done.

Response 9: We agree with you that removing some parts can make the whole manuscript more clear and definite. Thank you for your advice. We decided to delete this paragraph.

Comments 10: Line 237 coated?

Response 10: We are very sorry. Maybe that's not the right word to use. We made changes in the manuscript.

Line 235: dumped in

Comments 11: Line 255 Figure4 what does this show? Are these 10 mycelia that had been co-cultivated with Agro and in the middle a segment called wt that has not been co-cultivated? Please give better explanation in the legend. Then in the text describe the number of independent experiments that were done in this way and how many plates were done in each experiment and then the results of each experiment and the calculation of the average transformation frequency for use to compare the influence of the variables tested.

Response 11: Thank you for pointing this out. I/We agree with this comment. The Fig. 4 shows the transformation efficiency of A. tumefaciens carrying different vectors (pCAMBIA-1301 and pCAMBIA-SH-gpd). The middle WT represents the mycelium mass that has not been co-cultured with A. tumefaciens. As can be seen from the fig. 4, mycelium mass without co-culture cannot grow in PDA medium (hygromycin 4 μg/mL). Your suggestions can make our pictures better understood. We have explained WT in more detail.

Line 256: WT: The mycelium mass was not co-cultured with A. tumefaciens.

Transformation efficiency = number of positives of mycelium mass /number of germinated mycelium mass. Each experiment (carrying pCAMBIA-1301 and pCAMBIA-SH-gpd vectors) consisted of three groups, and the final result was the average of the 3 groups. In experiments, the number of mycelium mass is different in each petri dish.

Line 249: After hygromycin screening, mycelium mass were selected for PCR validation.

Comments 12: Line 261 pollution rate?

Response 12: We have revised this sentence.

Line 260: The transformed material was contaminated, which has a lot to do with the operation steps and the regeneration efficiency of the material.

Comments 13: Line 263 contamination of transformation material?

Response 13: The meaning here is that after the protoplast is prepared, the material was contaminated as the culture time is prolonged. In order to make the meaning of this sentence clearer, and we have revised the sentence.

Line 260: The transformed material was contaminated, which has a lot to do with the operation steps and the regeneration efficiency of the material.

Comments 14: Line 271 rate

Response 14: Our previous manuscript wasn't very clear. We have made changes.

Line 263: In this study, there was material contamination in protoplast treatment, and the material was not contaminated by using the other two treatments.

Comments 15: Line 314 bivector? Hydomycin

Response 15: This phrase means "vector carrying hygromycin screening marker". Perhaps our expression is not very clear, and we have revised this sentence.

Line 313: Since the p1301-SH-gpd vector carried a screening marker for resistance to hydomycin.

Comments 16: Line 379 the authors suggest that S.b. was now transformed for the first time, but they already published last year about it (reference 37 in this ms).

Response 16: Thank you very much for your suggestion, which makes our manuscript more rigorous. We have revised this sentence.

Line 271: This study successfully optimized the operation steps of genetic transformation for the first time.

4. Response to Comments on the Quality of English Language

Point 1: English in general is not too bad. Sometimes the wrong words are used as I have indicated specifically above.

Response 1: Thank you very much for your review of our manuscript. We have revised all the questions you raised, and we have written the sentences more clearly. Thank you very much.

Reviewer 3 Report

Comments and Suggestions for Authors

In this study, the limited research on the gene function of Sanghuangporus Baumii, a valuable medicinal fungus, is mentioned due to the absence of a genetic transformation system. The main objective was to establish an efficient transformation system mediated by Agrobacterium tumefaciens (ATMT) for S. Baumii. As a result, an efficient transformation system for S. Baumii, which opens new possibilities for genetic studies and biotechnological applications in this medicinal fungus.

It seems to me an interesting and relevant investigation for the Journal of Fungi, however, several modifications are required to be accepted.

Next, the changes to be considered:

Introduction

The introduction is satisfactory, however, it is required that Sanghuangporus baumii, its place of origin and the traditional use of this fungus and the main active components is described more taxonomically.

Materials and methods

Figure 1 should improve its quality

Line 137. Put S. Baumii in italics

Line 144, 149, 159 and 177. Reference of the methodology used

In section 2.5. Infection and co-cultural, the colonies selection criteria of A. Tumefaciens are not mentioned, please mention. The paragraph is also confusing, it is recommended to restructure it completely.

In section 2.6. In the screening of positive transformants it is necessary to mention the reaction times of the PCR.

In section 2.7. 2.7. Optimization of ATMT System for S. Baumii, it is necessary to mention or write down the formula of how the transformation rate was calculated.

Results

The results presented are clear, however, more evidence is required to complement the most significant of the study.

Add:

Number of strains transformed in Hygromycin to different concentrations and mention the ideal concentration for study.

Put an image of an not transformed strain to a high dose (6 µg/ml) of hygromycin.

Annex the PCR gel where expretion bands are displayed, since the study is mentioned that it was repeated twice, but the images are not shown.

It is recommended to carry out a graph of the kinetic of exprecion of protoplasts in the different conditions of the experiment and contrast the transformation rate.

Discussion

In line 349 it is mentioned that the efficiency of the procedure is improved, annexed as a percentage, the achievement obtained.

Conclusion

Adequate, however, fig. 6 must be eliminated from this section. It is recommended to change the image at the end of the discussion, with a brief section.

Round 2

Reviewer 1 Report

Comments and Suggestions for Authors

The manuscript is greatly improved

Comments on the Quality of English Language

The English has still to be improved.  It is not flowing, and unclear at some instances

Reviewer 2 Report

Comments and Suggestions for Authors

The authors have carefully addressed my critical points. The figures are also  now of a good quality.